# Novel multi-energy X-ray imaging methods: Experimental results of new image processing techniques to improve material separation in computed tomography and direct radiography

**Mirko Heckert**[ID]*, **Stefan Enghardt, Jürgen Bauch**

Institute of Materials Science, Technische Universität Dresden, Dresden, Germany

* mirko.heckert@tu-dresden.de

**Data Availability Statement:** Data are available from figshare: 10.6084/m9.figshare.11931726.

## Abstract

We present novel multi-energy X-ray imaging methods for direct radiography and computed tomography. The goal is to determine the contribution of thickness, mass density and atomic composition to the measured X-ray absorption in the sample. Algorithms have been developed by our own to calculate new X-ray images using data from an unlimited amount of scans/images of different tube voltages by pixelwise fitting of the detected gray levels. The resulting images then show a contrast that is influenced either by the atomic number of the elements in the sample (photoelectric interactions) or by the mass density (Compton scattering). For better visualization, those images can be combined to a color image where different materials can easily be distinguished. In the case of computed tomography no established true multi-energy methodology that does not require an energy sensitive detector is known to the authors. The existing dual-energy methods often yield noisy results that need spatial averaging for clear interpretation. The goal of the method presented here is to qualitatively calculate atomic number and mass density images without loosing resolution while reducing the noise by the use of more than two X-ray energies. The resulting images are generated without the need of calibration stan-dards in an automatic and fast data processing routine. They provide additional information that might be of special interest in cases like archaeology where the destruction of a sample to determine its composition is no option, but a increase in measurement time is of little concern.

## Introduction

Multi-energy radiography is known to have the potential to overcome certain disadvantages of conventional single-energy radiography [1, 2]. In the latter, the only available information is the detected gray level of a specific image region, which corresponds to the X-ray attenuation

**Funding:** M. H. and S. E. recieved funding of the Sächsische Aufbaubank and the European Union (EFRE) for the project MERL-F within the InfraPro initiative. The X-ray equipment used for the described experiments was funded by the project MERL-G from the same organization. The funders had no role in study design, data collection and analysis, decision to publish, or preparation of the manuscript.

**Competing interests:** The authors have declared that no competing interests exist.

in the sample. For a given X-ray energy it is dependent on the path length of the X-ray in the sample as well as the density and the elemental composition of the sample (atomic number). Many different combinations of these factors can generate the same gray level in the X-ray image, which might lead to a misinterpretation of the results (Fig 1).

As the probability of photon-matter interactions (here mainly Compton scattering and photoelectric effect) shows a unique energy dependence for each element, the use of several images recorded with different X-ray energies can allow for a better material separation. The most simple approach of multi-energy diagnostics is using only two distinct energies (or X-ray tube voltages). This allows for a limited categorization of the material into classes with a big difference in their absorption behavior, like organics, minerals and metals. This technique is widely used in baggage scanners. The usage of more than two X-ray energies may enable a better material separation as the transmission curves in Fig 1 show a different trend even for very similar materials like the polymers PS and PET, see also [5], [2].

Even under the assumption of a perfect material segmentation, in conventional radiography the density and the thickness information of the object cannot be separated. For this purpose a computed tomography (CT) of radiographs can be reconstructed using images taken from different angles. The multi-energy calculation can then be used either directly on the recorded projections or on the volumetric data from different kV values. In any case the 3 dimensional multi-energy CT data leads to challenges in computing and data handling.

First methods of dual-energy computed tomography have been described 1976 by Alvarez and Macovski [1]. In 2003 Heismann [6] introduced the $\rho Z$-projection that allows for the calculation of two new images from dual-energy CT data. One of those images shows a mass density dominated contrast, the other represents the atomic number. These methods have been proven to allow for a good material separation, and even for a quantitative measurement of $\rho$ and Z [7]. Investigations using more than two X-ray energies or energy bins in the case of a spectroscopic detector can be found in [8] [9] [10] [11].

A common issue of all this studies is that the presented methods often need an extensive calibration for every material system investigated. Furthermore image noise seems usually to be a problem [12] that is then countered by spatial averaging of the resulting data.

In this paper we present a method that allows for the calculation of a $\rho Z$-like projection in 3D without any sample specific calibration and with very little user interference (i.e. fast and automatic image processing, no filter change needed during measurement). It is not designed

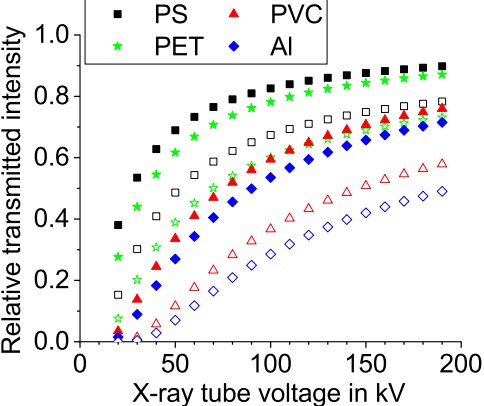

**Fig 1. Simulated X-ray transmission for different materials.** Open symbols represent a thickness of 5 mm, filled symbols 1 mm. The absorption coefficients are taken from [3] and the tube spectra have been simulated according to [4].

for absolute determination of $\rho$ and Z. The image noise is reduced by the usage of more than two CT scans so that theoretically no image resolution is lost. A computer program was written to extract the material information from a series of radiographs/tomographs taken at different kV values in a simple and universally applicable process. Color coded images can than be generated where materials are represented by different colors and thickness/density are represented by the shading.

## Theory

Radiographic images show a contrast dependent on the absorption coefficient $\mu$ and the path length $t$ of the X-ray beam in the sample. $\mu$ is a function of the sample density $\rho$, the atom number $Z$ and the X-ray energy $E$. For $E$ in the keV range of typical X-ray tubes $\mu$ consists of two main components, the photoelectric interactions $\tau \sim \rho Z^4 / E^3$ and the Compton scattering $\sigma \sim \rho Z \cdot f_{KN}(E)$ where $f_{KN}(E)$ denotes the Klein-Nishina function in the form of [1].

To distinguish materials based on several radiographs of different X-ray energies, new images are calculated that show atomic number- or thickness/density-contrast. This is done by analyzing the energy-dependent absorption curve for each pixel on the image.

The gray value visible on the detector may be described as:

$$V(E_{max}, t) = \int_0^{E_{max}} D(E) I_0(E) \exp(-\int_0^t \mu(E,x)(x)dx) dE \qquad \text{with } E_{max} = Ue \qquad (1)$$

Here $I_0$ denotes the spectrum of the X-ray source, $t$ the sample thickness, $U$ the X-ray tube voltage, $e$ the charge of the electron and $D$ is given by the quantum efficiency and bias of the detector. Theoretically one could fit Eq 1 to experimental data and the resulting curves of $\mu$ could be used to differentiate between multiple materials or material combinations. But typically the spectra $I_0$ and $D$ are not known with sufficient precision to make such a fit feasible. Therefore approximations have to be used. The motivation of our evaluation method can be visualized by setting $D = 1$ and replacing $I_0$ with Kramers' law [4]. This strong simplification ignores the characteristic radiation. But the relative intensity of a single X-ray beam attenuated by a homogeneous sample then becomes simply:

$$V_{rel}(E_{max}, t) = \frac{V(E_{max}, t)}{V(0)} \approx \frac{\int_0^{E_{max}} K\left(\frac{E_{max}}{E} - 1\right) E \exp(-\mu(E)t) dE}{\int_0^{E_{max}} K\left(\frac{E_{max}}{E} - 1\right) E \, dE} \qquad (2)$$

$$V_{rel}(E_{max}, t) \approx \frac{2}{E_{max}^2} \int_0^{E_{max}} \left(\frac{E_{max}}{E} - 1\right) E \exp(-\mu(E)t) dE \qquad (3)$$

The diagram in Fig 1 shows the comparison between 4 materials of two different thicknesses each when using the attenuation data of [3]. The difference in curvature of the graphs demonstrates the variable influence of the attenuation coefficients of the simulated materials.

## Materials and methods

All experiments have been performed on a diondo d2 CT/CL-System with a maximum focus-detector-distance of 1167 mm. The X-ray tube (X-ray-works XWT-190-CT with Cu-target) can be operated from 20–190 kV with a current of 0.05–3 mA and has a JIMA-resolution of up to 2 $\mu$m. The tube window is made of beryllium and no additional filters have been used. The Detector is a flat panel (Varex PaxScan 4343DX-I) with 3072 x 3072 px$^2$ and 139 $\mu$m pixel pitch. The cover plate of the active area is made of carbon fiber composite which enables a good signal even at lower X-ray energies.

**Table 1. Material properties of the samples investigated in this paper.**

| Material | $Z_{eff}$ | $\rho$ in $g/cm^3$ | thickness in $mm$ | $\mu/\rho_{100kV}$ in $cm^2/g$ |
|---|---|---|---|---|
| PVC | 14.3 | 1.33 | 0.5–15.7 | 0.1887 |
| PS | 5.7 | 1.04 | 0.5–16.6 | 0.1624 |
| PET | 6.7 | 1.38 | 0.6–18.2 | 0.1586 |
| Silicone | 10.6 | 1.14 | 0.5–17.7 | 0.1760 |
| Aluminum | 13 | 2.7 | 0.3–5.9 | 0.1704 |
| Paper | 6.9+x | 0.73 | 0.5–8.0 | 0.1621+x |
| Sugar | 7.0 | 1.07 | 5.5–16 | 0.1626 |
| PTFE | 8.5 | 2.2 | 0.1–0.5 | 0.1500 |
| Glass | 12.9 | 2.5 | fiber in epoxy | 0.1740 |

$Z_{eff}$ calculated according to [13], $\mu/\rho$ values derived from [3]. For the paper sample the +x denotes the contribution of the $CaCO_3$ used as filler.

Samples with stepwise varied thicknesses were used, their material properties and the thickness range investigated can be found in Table 1. Since the exact composition of the paper sample is unknown, the effective atom number $Z_{eff}$ and the mass absorption coefficient $\mu/\rho$ have been calculated for pure cellulose. The real values for $Z_{eff}$ and $\mu/\rho$ are higher due to the calcium carbonate used as filler. For the polymer (PVC, PS and PET), silicone and paper sample a exponentially increasing step-thickness was used. The aluminum sample has a linearly growing thickness due to the higher absorption of X-rays.

To extract the spectroscopic information from the acquired multi-energy image series a pixelwise fitting of the transmitted intensity with respect to the tube voltage has been conducted. This may be done using a numerical approximation of Eq 3 of the form:

$$V(U, t) = K \cdot \Delta U \cdot \sum_{i=1}^{n} \left( \frac{U}{i\Delta U} - 1 \right) i\Delta U \exp(-\mu(i\Delta Ue)t) \tag{4}$$

Here the integral is replaced by a sum where the spectrum and attenuation function is evaluated at discrete energies $i\Delta Ue$.

Eq 4 leads to a linear equation system of the variables $\exp(-\mu(i\Delta Ue)t)$ with respect to the tube voltage steps $U_{1,2,\ldots n}$. This enables a relatively quick solution of the integral in Eq 1.

The terms $\mu(i\Delta Ue)t$ can than be replaced by $\mu \approx a_{ni}E^{b_{ni}} + c_{ni}f_{KN}(E) + d_{ni}$ similar to the approach of [1] so that the full equation reads:

$$V(U) = \Delta U \cdot \sum_{i=1}^{n} \left( \frac{U}{i\Delta U} - 1 \right) i\Delta U \exp\left( a_{ni}E^{b_{ni}} + c_{ni}f_{KN}(E) + d_{ni} \right) \tag{5}$$

$a_{ni}$ and $c_{ni}$ contain information about the thickness as well as the atomic number of the sample while $d_{ni}$ is determined by the thickness as well as device parameters and constants (e.g. Kramers' constant $K$). $c_{ni}$ also shows a small dependence on the material but should roughly be equal to -3 [1]. This leads to a nonlinear fit that is still relatively slow compared to other fitting methods. For numerical stability reasons the exponent $b_{ni}$ is fixed to -3.

In order to perform this fit the input data needs to be free of (energy dependent) influences of the experimental setup. As described in the results section, the detector used here has a lower quantum detection efficiency for higher X-ray energies. This leads to interpretation problems of the resulting fit parameters $a_{ni}$, $c_{ni}$, and $d_{ni}$ especially because the device parameter

contained in $d_{ni}$ shows a presently unknown energy dependence that also influences the other parameters.

Another approach to avoid a biased result is the fitting of the multi-energy image data with empirically obtained functions. For direct radiographs the fitting can be done with a 3rd order polynomial function of the form $V(U) = a_{dr}U^3 + b_{dr}U^2 + c_{dr}U + d_{dr}$. Thickness and material information need than to be extracted from the fitting parameters $a_{dr}$–$d_{dr}$. Prior to this all acquired images were bias- and dark-field corrected by the measurement software with correction images recorded at 110 kV and then divided by the tube current, the integration time of the detector and for the polynomial fitting the square of the high voltage to compensate the different measurement conditions. After this normalization the measured intensities (see Results section) behave roughly like the calculated intensities in Fig 1 with a drop of the measured intensities at high voltages above 120 kV due to a lower sensitivity of the detector for harder X-rays. It is planned to account for this behavior computationally in order to gain better results with the approach of [1] in a further improvement of the method.

The normalization of the input images and the fitting has been done with a newly developed computer program MultiE-GUI written in C++ (available for download at https://figshare.com/articles/MultiE-GUI_zip/9758390/1 including the mostly uncommented source code). It reads the integration time, current and tube voltage from the filename of the input files and is capable of batch processing for image stacks. The calculated fitting parameters are then again saved as gray levels (16 bit integer, automatically normalized) or as 64 bit floats for the corresponding pixels in a number of image files (either.tif or.raw). It allows for the use of all fit functions described in this paper including Eq 5 (called "Fit Numeric Integral HQ") as well as some other simple functions.

The resulting parameters of the 3rd order polynomial fit can be combined to a color image by assigning the sum of parameters $a_{dr}$ and $d_{dr}$ to the red channel, parameter $c_{dr}$ to green and $d_{dr}$ to blue. Depending on the sample and measurement conditions, some of this parameters have to be inverted so that they show bright objects on dark background before combining them to a color image. This image processing has been performed with ImageJ [14].

For multi-energy computed tomography the procedure is similar. CT-scans for various energies are conducted and reconstructed with a fixed scaling factor. The reconstructed slices are then sequentially processed with the MultiE-GUI program. The reconstructed slices show not only an inverted contrast compared to the projections, also the energy dependence of the gray values from the reconstructed slices is different (see Fig 2). The background is for all tube voltages almost constant and the gray level of regions inside the sample generally decreases with growing tube voltage. To fit this data, no further normalization is needed as the CT reconstruction already sets a fixed background. To generate a material contrast image an Arrhenius plot is created (log gray level vs 1/high voltage). The curves then represent almost perfect parabolas and may be described as a function of the form $W(U) = e^{a_{ct}U^{-2} + b_{ct}U^{-1} + c_{ct}}$ where W(U) denotes the gray level after CT reconstruction. For CT it is necessary to chose 64 bit float raw images as output format of the fit, as the normalization of each output image (slice) would lead to an unsteady background. The raw image stacks are than imported in ImageJ and converted to 16 bit grayscale. While opening the image stack the gray levels are automatically normalized for highest and lowest float values of the whole stack. The parameters can then again be mapped to colors $a_{ct}$ to blue, $c_{ct}$ to yellow) leading to a RGB image stack that can be rendered in 3D, for example with the open source software ParaView [15].

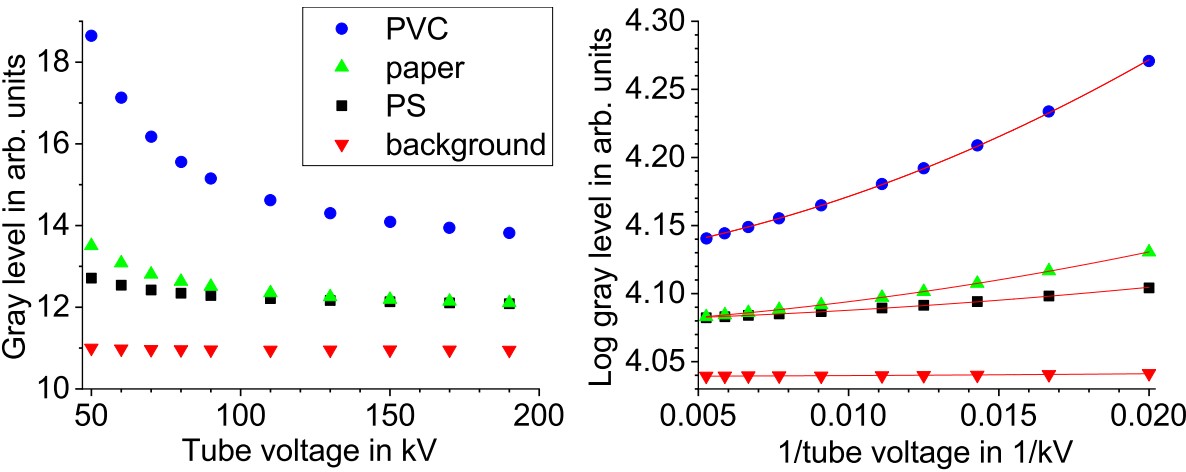

**Fig 2. Gray value vs X-ray tube voltage after CT reconstruction.** The gray values have been extracted from reconstructed CT slices at different regions of the sample shown in Fig 8. This corresponds to different materials. The data in the right plot is fitted with:
$$W(U) = e^{a_{ct} U^{-2} + 0.01 a_{ct} U^{-1} + c_{ct}}.$$

## Results and discussion

### Direct radiography

Radiographs of step wedge samples made of PS, PVC, PET, aluminum, paper and silicone have been recorded at 40–190 kV X-ray tube voltage. The measured intensities for some image regions can be found in Fig 3. Below 40 kV the transmission through the sample was too low to reach a linear detector behavior.

These images have been used for a 3rd degree polynomial fit of the intensities with the program MultiE-GUI. The resulting parameter images are shown in Fig 4. They do not directly represent any material or radiographic properties, but it can already be seen that the parameters $a_{dr}$, $b_{dr}$ and $c_{dr}$ behave roughly like the radiographs (see Fig 5 left image) themselves (some of them inverted) and that the parameter $d_{dr}$ shows a somewhat different behavior

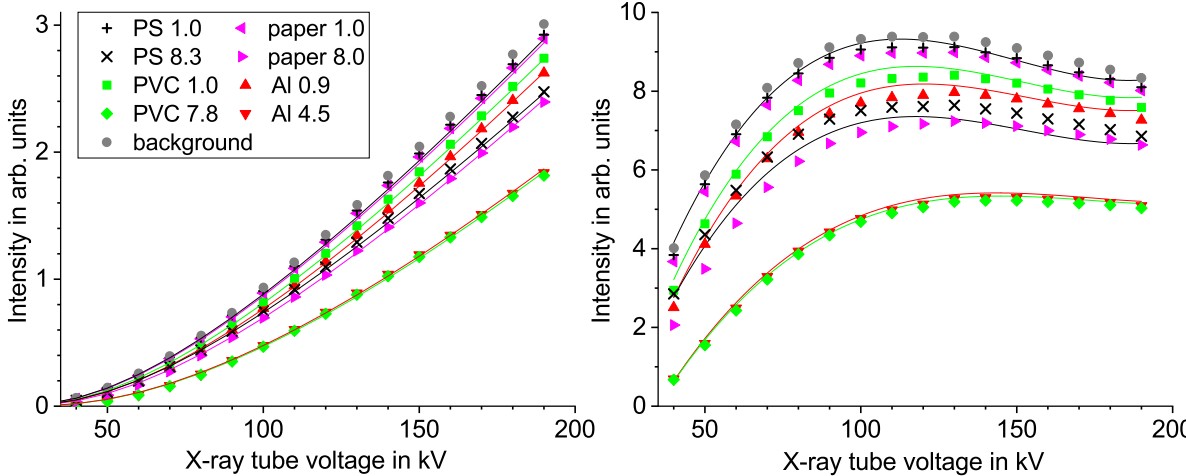

**Fig 3. Intensity vs X-ray tube voltage for direct radiography.** Gray levels of PS, paper, PVC and aluminum samples at different thicknesses (in mm) and the background. Data in the left plot is normalized with tube current and detector integration time fitted with Eq 5. In the right plot the data is additionally normalized with the square of the tube voltage and fitted with a cubic function.

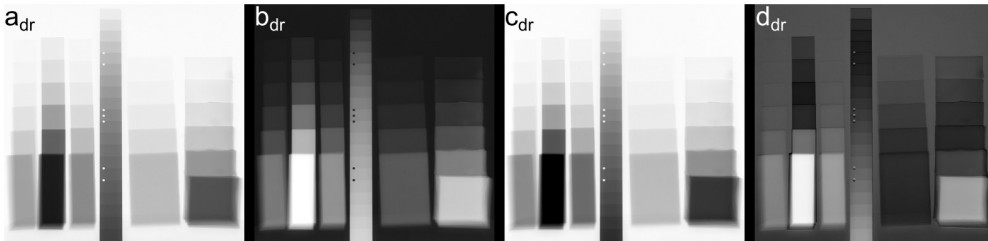

**Fig 4. Multi-energy fit parameters represented as gray levels.** Four coefficients ($a_{dr}$, $b_{dr}$, $c_{dr}$, $d_{dr}$) of a 3rd degree polynomial fit of a multi-energy image series of PS, PVC, PET, aluminum, paper and silicone from 40-190 kV.

emphasizing the material differences, but still the contrast is mainly dominated by the thickness, with an inversion for thicker samples containing heavier elements.

An image with a material emphasized contrast where the gray level still follows (mostly) the thickness can be generated by plotting the coefficients $c_{dr} + d_{dr}$, see Fig 5B. The color representation in Fig 5C has been generated from the fit coefficients by mapping $c_{dr} + d_{dr}$ inverted to the red, $b_{dr}$ to the green and $d_{dr}$ to the blue color channel. On that image, the material groups are colored with a similar hue and the brightness is a function of the sample thickness.

This method allows for a (limited) extraction of the material information from conventional radiographs of different X-ray energies with a reduction of the influence of the sample thickness in the resulting image. Still, a clear classification of the material groups is only possible for a limited thickness range. To overcome this disadvantage a multi-energy computed tomography can be performed.

For comparison, the images from 50 kV and 100 kV have been used to calculate a dual-energy image using the logarithmic transparency quotient (see [16]). The results are presented in Fig 6 as grayscale and color coded image.

It can be seen that for direct radiography the multi-energy method gives no clear advantage compared to the dual-energy method. That is most likely due to the fact that the unknown detector influence on the measured intensities is less affecting the logarithmic transparency coefficient than the polynomial fit. As can be seen in Fig 3 the curves for the 1 mm PS sample behave almost identical to the background, therefore the multi-energy fitting is dominated by the instrument influence and not the sample. For low energies the detector window (5 mm

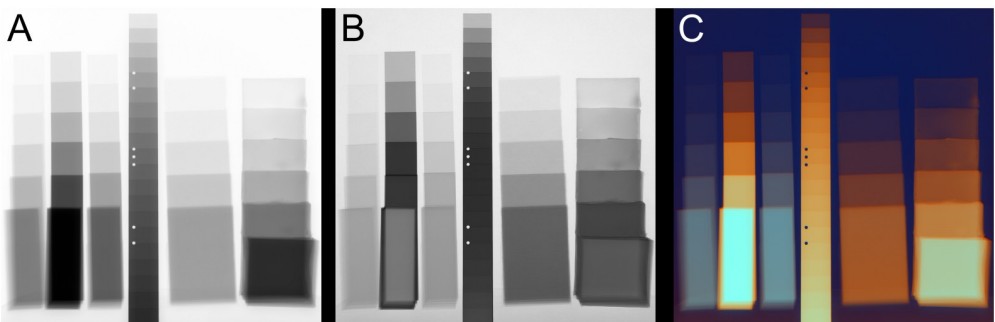

**Fig 5. Comparison of single- and multi-energy radiographs.** A: Conventional radiography of PS, PVC, PET, aluminum, paper and silicone samples at 100 kV showing thickness-dominated contrast; B: multi-energy image (coefficients $c_{dr} + d_{dr}$) showing material contrast; C: color representation of the calculated coefficients showing material and thickness contrast.

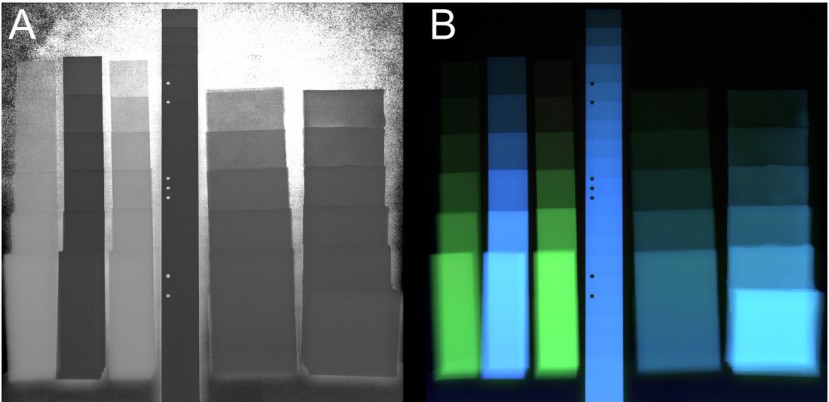

**Fig 6. Dual-energy calculation (logarithmic transparency quotient) from the 50 kV and 100 kV radiographs.** A: grayscale output of the dual-energy calculation. B: colored representation using the dual-energy data as hue and saturation channel and the 100 kV image as intensity.

carbon fiber) has a big influence, for higher energies the quantum efficiency seems to degrade as the normalized intensities decrease above 130–150 kV.

Additionally in Fig 7 the results of a fit of Eq 5 to the multi-energy radiographs discussed above are presented. The parameter $b_{ni}$ is fixed to −3 and is therefore not shown. Except for the thickest step of PVC the fit yielded reasonable results (raw data can be downloaded from figshare, see the data availability statement) and a relatively good approximation of the data points (see also Fig 3). Nevertheless, no material information could be extracted from this results. The energy dependent detector efficiency has an influence on all fitting parameters overlaying their material dependent behavior with (currently) unknown device characteristics.

## Multi-energy CT

The demonstration sample for multi-energy computed tomography consists of the PVC, PET and paper step wedge samples from the previous section, together with sugar cubes, 3 stripes of basalt- glass- and polyester-fibers in epoxy, bound together with PTFE-tape, see Fig 8. Further results are shown for mineral samples and 2 light bulbs (see S3 Video) as prototype of real samples. We didn't use classical phantoms as they mainly represent the situation of a medical CT. The engineering samples we investigate often consist of materials of different absorption separated by large air filled volumes. This leads to far more reconstruction artifacts. However,

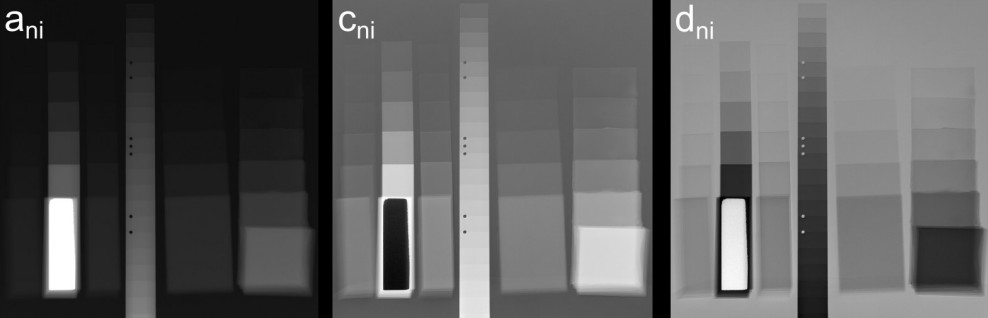

**Fig 7. Results of the numeric fit of the integral in Eq 1.** Fitting parameters of the term $\mu \approx a_{ni}E^{b_{ni}} + c_{ni}f_{KN}(E) + d_{ni}$ represented as grayscale images. Parameter $b_{ni}$ is fixed to −3.

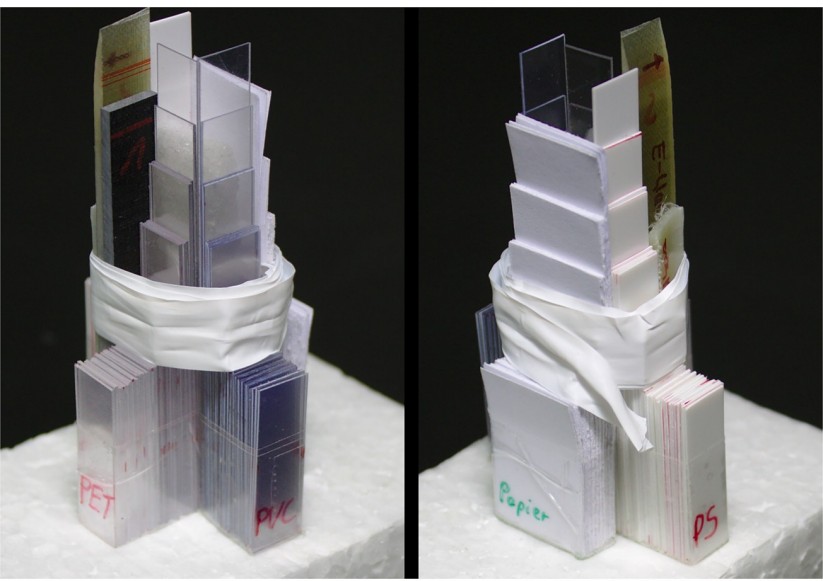

**Fig 8. Sample for multi-energy CT.** PET, PS, PVC and paper wedge samples with strips of glass-, basalt- and polyester-fiber in epoxy around sugar cubes wrapped together with PTFE-tape.

it was the goal of this investigation to develop a robust multi-energy method to cope with this difficulties.

10 CT-scans of the demonstration sample have been conducted at a focus object distance of 200 mm and a focus detector distance of 700 mm, at 50, 60, 70, 80, 90, 110, 130, 150, 170, 190 kV, 900–130 $\mu A$, 2–0.6 s integration time. The filtered back projection reconstruction was performed with the Siemens CERA software. The gray levels for some regions of the reconstructed slices are presented in Fig 2. Cross sections of a conventional CT reconstruction for projections recorded at 90 kV X-ray tube voltage can be seen in Fig 9. There, the PVC, the basalt- and glass-fibers can clearly be separated from the other materials, but the paper has a very similar gray level to the lighter materials. Furthermore beam hardening artifacts can be seen in all materials that would affect a material separation based on gray level thresholding.

As mentioned above, a exponential fit of the form $W(U) = e^{a_{ct}U^{-2} + b_{ct}U^{-1} + c_{ct}}$ was then used to extract the spectroscopic information from the reconstructed CT slices. That leads to a very good approximation with a $R^2$ of 99,993%–99,996% for the curves of the 3 materials in Fig 2, but the data of the parameters $a_{ct}$ and $b_{ct}$ was extremely noisy, as this two parameters interfered with each other. From the fit functions of various multi-energy data the correlation $b_{ct} = 0.01a_{ct}$ was derived. The final fit function then reads $W(U) = e^{a_{ct}U^{-2} + 0.01a_{ct}U^{-1} + c_{ct}}$. This reduced the noise to an acceptable level while the $R^2$ decreased only slightly to 99,909%–99,995% for the data discussed above. Furthermore, it allows for a reduction of the acquired X-ray energy steps.

The resulting images of the exponential fit of the multi-energy CT scans are shown in Fig 10. Parameter $a_{ct}$ (Fig 10A) shows a pronounced Z-contrast where materials with higher atomic numbers are brighter. The parameter $c_{ct}$ (Fig 10B) behaves like a standard CT reconstruction showing a density-dominated contrast. It's worth mentioning that the beam hardening artifacts are greatly reduced in the picture Fig 10B, even compared to a 190 kV scan.

In Fig 11 colored cross sections of the multi-energy CT reconstruction can be found that were calculated mapping parameter $a_{ct}$ to the blue channel and parameter $c_{ct}$ to yellow. They

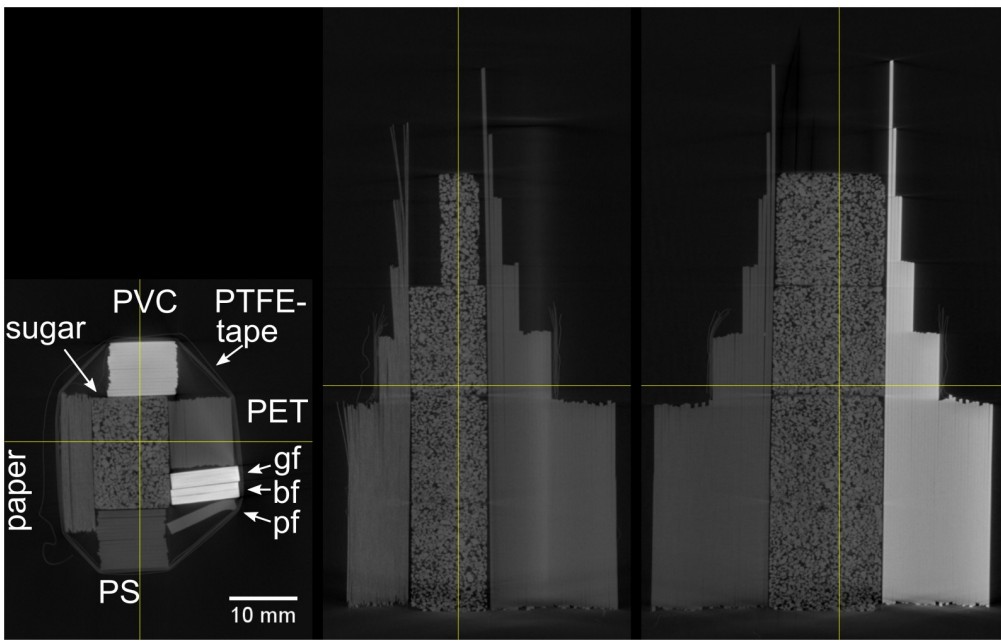

**Fig 9. Cross sections of a conventional CT reconstruction of the sample shown in Fig 8.** Recorded at 90 kV, abbreviations of sample materials represent gf: glass fiber; bf: basalt fiber; pf: polyester fiber.

show a material separation by color coding, sugar, PET and polyester-fiber have the same greenish color because they consist mainly of the same elements. Here, a clear separation of the paper from those elements is possible, showing the presence of heavier elements (fillers) in the paper. PVC and the fiberglass samples show individual colors as well.

From the multi-energy CT slices a 3D representation can be calculated. This has been done with the program ParaView [15] as many conventional CT image analysis programs do not support RGB image stacks. The transparency is determined by the green channel. An 3D image of the multi-energy CT result can be found in Fig 12 with a reconstruction of the 90 kV

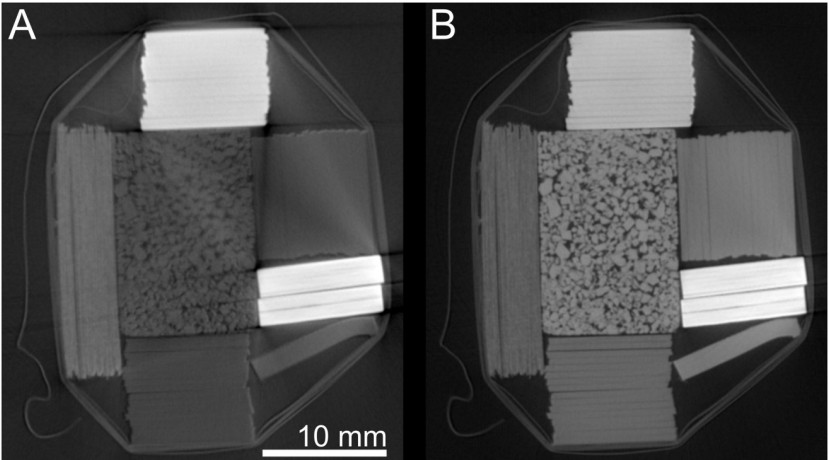

**Fig 10. Resulting grayscale images from a multi-energy reconstruction.** Parameters from the fit function $W(U) = e^{a_{ct}U^{-2}+0.01a_{ct}U^{-1}+c_{ct}}$ represented as gray levels. A: parameter $a_{ct}$, B: parameter $c_{ct}$.

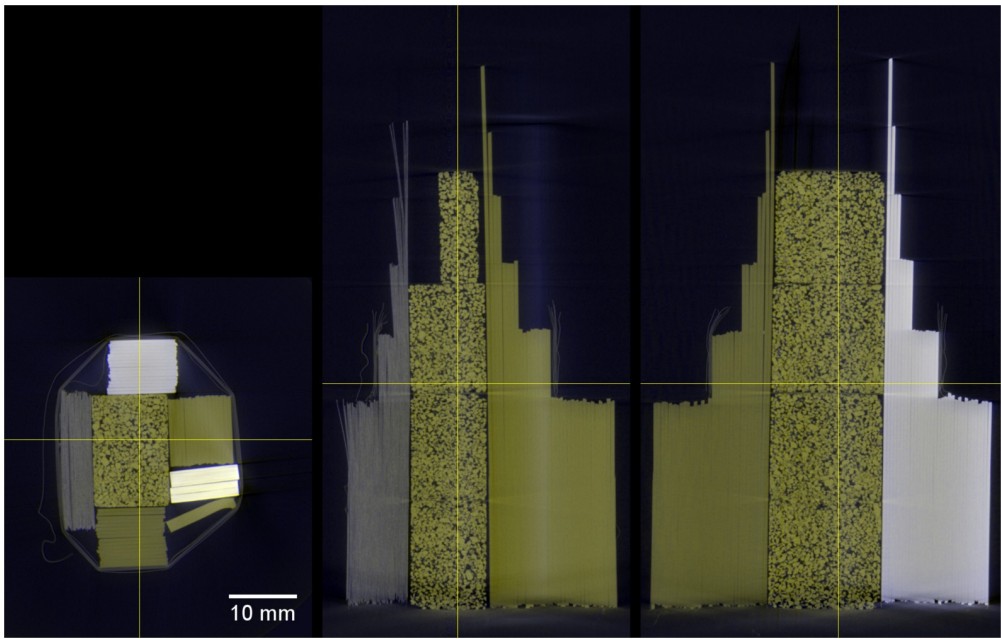

**Fig 11. Cross sections of a multi-energy CT reconstruction of the sample shown in Fig 8.** Parameter $a_{ct}$ is mapped as blue and $c_{ct}$ as yellow.

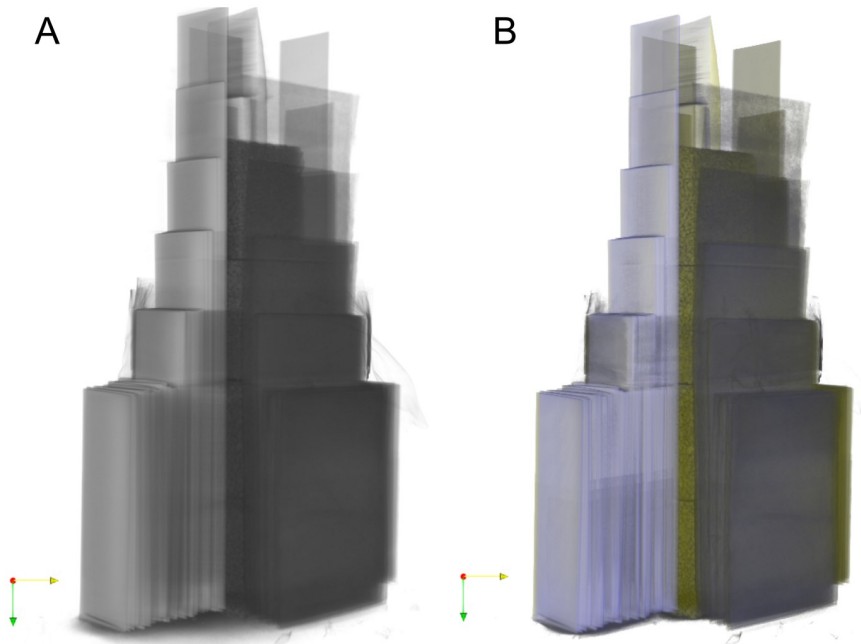

**Fig 12. 3D representations of the samples shown in Fig 8.** A: single-energy reconstruction at 90 kV. B: colored multi-energy reconstruction.

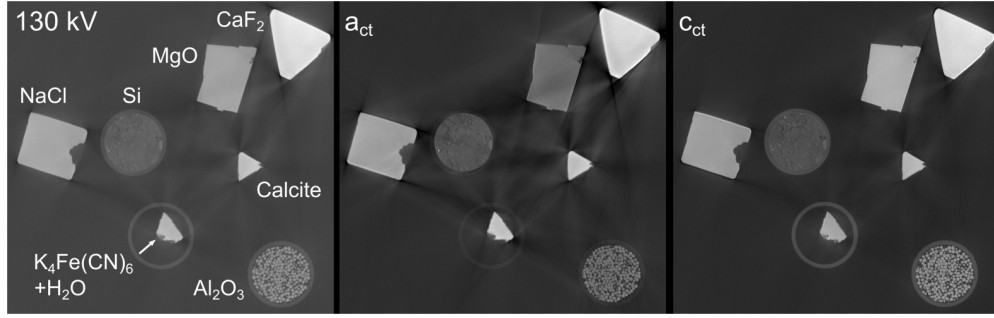

**Fig 13. Slices from a single- and multi-energy reconstruction of mineral samples.** Measured with a 2.5 mm Al filter.

CT scan for comparison (see S1 Video for a video of the 3D data). For multi-energy CT a clear distinction of the different materials is possible.

Streak-like artifacts in the material are still present in the multi-energy reconstruction, but they affect mainly the brightness and not the hue of a pixel so that a material classification is still possible. Beam hardening artifacts around samples have a distinctive color that is not assigned to any material as can be seen around the PVC sample in the 3D rendering in Fig 12. In conventional CT these artifacts have a gray level that is similar to the lighter sample materials for example the sugar. Without prior knowledge of the sample geometry it is often not possible to tell if it is an artifact in conventional CT.

Additionally, several minerals have been investigated with the multi-energy CT method. The bigger pieces were supported on expanded polystyrene, the smaller crystals as well as the silicone powder and the alumina pellets are contained in PP centrifuge tubes. For one slice the resulting parameters $a_{ct}$ and $c_{ct}$ as well as a conventional reconstruction at 130 kV are mapped to gray levels in Fig 13.

The same slice as colored multi-energy reconstruction can be found together with 3D renderings in Fig 14, for a video of the colored 3D representation see S2 Video. The colored multi-energy data enables a far better material separation in comparison to the single-energy CT-reconstruction. The yellow and blue rim best visible at the $CaF_2$ sample is caused by a slight misorientation of the scans at different kV values caused by position uncertainties of the sample manipulator and a shift of the beam center when changing the tube voltage. This

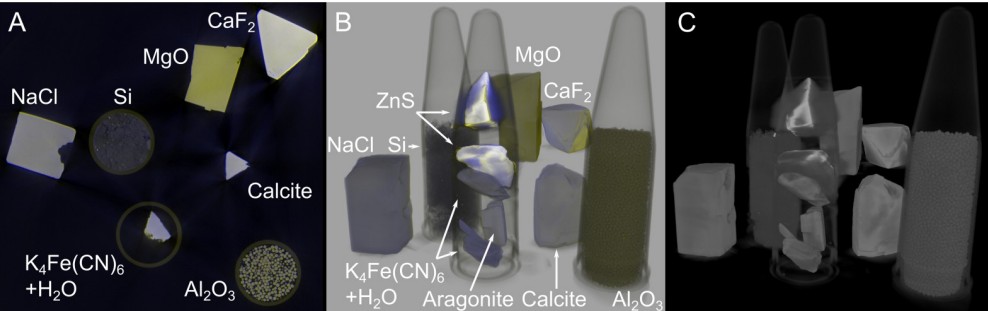

**Fig 14.** Slice (A) and 3D rendering (B) of a multi-energy reconstruction of the mineral samples with single-energy grayscale rendering (C) for comparison. Measured with a 2.5 mm Al filter, grayscale 3D rendering with scatter renderer of the software VG-Studio [17] (130 kV), colored multi-energy rendering with the software ParaView [15].

**Table 2. Material properties of the samples shown in Fig 13 and values of the fitting parameters $a_{ct}$ and $c_{ct}$ from the multi-energy CT measurement.**

| Material | $Z_{eff}$ | $\rho$ in $g/cm^3$ | $a_{ct}$ | $c_{ct}$ |
|---|---|---|---|---|
| PP | 5.5 | 0.91 | 36 | -2.274 |
| MgO | 10.8 | 3.58 | 169 | -2.186 |
| Corundum ($Al_2O_3$) | 11.3 | 4 | 219 | -2.215 |
| Silicon powder | 14 | 1.2 | 133 | -2.267 |
| NaCl | 15.3 | 2.16 | 357 | -2.2 |
| Calcite ($CaCO_3$) | 15.6 | 2.73 | 439 | -2.179 |
| Aragonite ($CaCO_3$) | 15.6 | 2.95 | 459 | -2.173 |
| $CaF_2$ | 16.9 | 3.18 | 512 | -2.153 |
| ($K_4Fe(CN)_6$) | 17.8 | 1.85 | 450 | -2.195 |
| ZnS | 26.1 | 4.1 | 590–1165 | -2.01—1.93 |

$Z_{eff}$ calculated according to [13].

sample was measured at 60, 70, 80, 90, 110, 130, 160 and 190 kV. Further measurement parameters can be found within the uploaded dataset at figshare, see the data availability statement.

The absolute values of the parameters $a_{ct}$ and $c_{ct}$ may be found in Table 2. It can be seen that parameter $a_{ct}$ follows the $Z_{eff}$ values mainly for samples with similar density. The sample applies for the parameter $c_{ct}$ and the sample density for similar $Z_{eff}$ values. For materials with a big difference in mass density and/or effective atom number the correlation with $c_{ct}$ and $a_{ct}$ respectively is not clear anymore. Nevertheless the information gained with the multi-energy CT is helpful for material separation in cases where different materials need to be told apart that may appear the same in single-energy CT and to recognize the same material inside similar samples at different locations. For heavy materials such as ZnS the parameter $a_{ct}$ changes significantly from surface to the bulk of the crystal. In that case it would be necessary to measure at higher tube voltages.

## Conclusion

Novel multi-energy image reconstruction methods have been presented and evaluated for direct radiography and computed tomography.

For direct radiography the goal was to separate the thickness/mass density and compositional contribution to the X-ray absorption in order to generate an image with increased Z contrast. However, even with applied normalization the measured intensity curves of the used tomography system were not behaving as expected but showing a decrease of the normalized intensity at higher kV-values. Therefore the results of a multi-energy image reconstruction based on the absorption integral cannot be used for material separation. A polynomial fitting of the multi-energy intensity data proved the possibility to analyze this data with a pixelwise fitting but showed no advantage over dual-energy methods.

For computed tomography, the published dual- or multi-energy methodologies often require a extensive calibration with standards similar to the investigated samples. With a more empirical approach it was discovered that the gray level of a volumetric image series can be fitted for each voxel with a function of the form $W(U) = e^{a_{ct}U^{-2}+0.01a_{ct}U^{-1}+c_{ct}}$. Parameter $a_{ct}$ is dominated by the Z-contrast and parameter $c_{ct}$ represents the mass density. From this data a color coded image may be generated where the different materials can easily be separated.

With this approach it might not be possible to get quantitative results, but it enables the investigation of a wide range of materials from polymers to metals. For the investigation of

metals and other materials with high absorption, beam filters may be used without changing the reconstruction procedure (see Results with Al-filter in Fig 13). Furthermore, no filter change is required during the multi-energy measurement and a noise reduction is accomplished by using more than two energy steps enabling the full resolution in the multi-energy output image.

## Supporting information

**S1 Video. 3D rendering of wedge shaped samples.** 3D representation of the multi-energy measurement shown in Fig 11 rendered with ParaView [15].
(MP4)

**S2 Video. 3D rendering of minerals.** Colored multi-energy CT reconstruction of the minerals in Table 2 measured from 60–190 kV with 2.5 mm Al-Filter rendered with ParaView [15].
(MP4)

**S3 Video. 3D rendering of 2 light bulbs.** Colored multi-energy CT reconstruction of 2 light bulbs (conventional and LED) measured from 70–190 kV with 2.5 mm Al-Filter rendered with ParaView [15].
(MP4)

## Acknowledgments

We thank Mandy Liebschner for the preparation of the silicone sample.

## Author Contributions

**Data curation:** Mirko Heckert.

**Formal analysis:** Mirko Heckert, Stefan Enghardt, Jürgen Bauch.

**Investigation:** Mirko Heckert, Stefan Enghardt.

**Methodology:** Mirko Heckert, Stefan Enghardt, Jürgen Bauch.

**Software:** Stefan Enghardt.

**Supervision:** Jürgen Bauch.

**Visualization:** Mirko Heckert.

**Writing – original draft:** Mirko Heckert, Stefan Enghardt.

**Writing – review & editing:** Stefan Enghardt, Jürgen Bauch.

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
