## [Decision Letter · Decision Letter 0]

27 Jan 2020

PONE-D-19-24675

Novel multi-energy X-ray imaging methods: Experimental results of new image processing techniques to improve material separation in computed tomography and direct radiography

PLOS ONE

Dear Mr Heckert,

Thank you for submitting your manuscript to PLOS ONE. After careful consideration, we feel that it has merit but does not fully meet PLOS ONE’s publication criteria as it currently stands. Therefore, we invite you to submit a revised version of the manuscript that addresses the points raised during the review process.

We would appreciate receiving your revised manuscript by Mar 12 2020 11:59PM. To enhance the reproducibility of your results, we recommend that if applicable you deposit your laboratory protocols in protocols.io, where a protocol can be assigned its own identifier (DOI) such that it can be cited independently in the future. For instructions see: http://journals.plos.org/plosone/s/submission-guidelines#loc-laboratory-protocols

We look forward to receiving your revised manuscript.

Kind regards,

David Fyhrie

Academic Editor

PLOS ONE

2. It is unclear why you have selected 'No - some restrictions will apply', 'For the multi-energy computed tomography data it may not be possible to share the whole dataset because of limited space (its around 500-800 Gbs). Some exemplaric slices and a scaled down dataset will also be made available via figshare.' Please clarify the nature of these restrictions.

PLOS only allows data to be available upon request if there are legal or ethical restrictions on sharing data publicly. For more information on unacceptable data access restrictions, please see http://journals.plos.org/plosone/s/data-availability#loc-unacceptable-data-access-restrictions.

Reviewers' comments:

Reviewer's Responses to Questions

**Comments to the Author**

1. Is the manuscript technically sound, and do the data support the conclusions?

Reviewer #1: Yes

2. Has the statistical analysis been performed appropriately and rigorously? 

Reviewer #1: Yes

3. Have the authors made all data underlying the findings in their manuscript fully available?

Reviewer #1: Yes

4. Is the manuscript presented in an intelligible fashion and written in standard English?

Reviewer #1: Yes

5. Review Comments to the Author

Reviewer #1: The authors introduce an empirical approach to perform a pixelwise fitting of the grey level of both radiographic images and a series of volumetric images corresponding to a CT scan measured with different X-ray tube voltages. The radiographs are fitted using a cubic interpolation while the CT scans are fitted using an exponential function. I believe the results are very interesting and depict a new method for empirical material differentiation on CT scans using conventional hardware and multiple X-ray energies. Therefore, I suggest its publication after the changes specified below. These are mostly required to justify the effectiveness of the approach, as well as to improve the scientific and technical rigor of the paper. Major points: - I do not think the relationship between the proposed fitting and the X-ray imaging model is clear. This mathematical relationship is, in my opinion, critical for replication of the results in the case of other materials. Since the authors claim the method to be qualitative rather than quantitative, my suggestion is for the authors to use a phantom with other materials to show the empirical method can do material differentiation in other cases too (Not as supplemental material with video 2 but as part of the discussion). - Related to the previous comment, further discussion is needed on the selection of the cubic interpolation for the radiography case. Furthermore, how did the authors get to the conclusion on how to use \\lambda, \\gamma and \\delta to generate the material images? The relation with the physical quantities is not clear from the theory developed in the paper. If the authors have explained this in a previous publication, please use a reference (line 125). - The notation is confusing. The authors do a nice work getting to equation (3) but then they go from V(t) to V(U) in equation (4) without an explanation and say it’s a derivation. This needs to be explained. Furthermore, V(U) is supposed to be the measured intensity at each tube voltage U, however, they claim to do a pixel/voxel grey level fitting for the CT reconstruction using the expression V(U) too. The latter is very confusing since this gray level corresponds to the attenuation coefficient quantities reconstructed for each voltage, these are not intensities. Please change the notation or clarify this accordingly. - Related to the previous comment, please explain why the second fit is on the linear attenuation coefficients rather than on the intensities as derived in equation (4). - Seems to me from the manuscript that the user needs prior information of the materials present in the sample to be able to identify which one corresponds to each parameter fitted. Is this correct? If not, please clarify How can one know which material is in the sample if the \\rho \\Z images are not quantitatively correct? - I believe there is not enough detail on the fitting to replicate the results of the paper. The authors do not give information on the type of software written, size of images, number of voltages, size of the step in the tube voltage, selection of n, size of the sample. Among others. Please elaborate, this information is often key for the reconstructions and cannot be derived from the manuscript. Minor points: - In the abstract the authors claim there are not stablished multi-energy methods. What about photon counting detectors? The statement needs to be revised. Maybe clarify what they mean with multi-energy method. On the other hand, to my knowledge there is one paper that relies on photon counting detectors to attain \\rho Z images [**] that is close a multi-energy method. Please elaborate. - In equation (1), e (which I assume is the charge of the electron) is not defined.

- In equation (4), K is not defined. Previous K in equation (2) came from the Kramer’s law but with the simplification in equation (3), K is not present anymore. Is the K in equation (4) different? Please, define. - The link for the fitting program MultiE-GUI does not work. I was able to find a software with the same name that seems to correspond to the description in the text, URL: https://figshare.com/articles/MultiE-GUI_zip/9758390/1. Please fix this. - If I understand correctly the approach in reference [1] is not used to fit /mu due to the lack of correction of the quantum efficiency at higher X-ray energies. Could you show the results that you get when you try performing the fit with your data? From what I saw on the .zip that I downloaded you have the option of performing this fit. I think this discussion will help the readers see a clear motivation of the proposed approach. - There are some typos along the paper. Please revise carefully before resubmitting. Some examples below: o Line 34: that instead of than o Line 112: change to: account for this behavior computationally o Line 182: An image instead of a image o Line 206: To an acceptable instead of to a acceptable o Line 249: A more instead of an more o Line 251: parameter instead of paramter [**] Busi, Matteo & Kehres, Jan & Khalil, Mohamad & Olsen, Ulrik. (2019). Effective atomic number and electron density determination using spectral x-ray CT. 2. 10.1117/12.2519851.

6. PLOS authors have the option to publish the peer review history of their article (what does this mean?). If published, this will include your full peer review and any attached files.

Reviewer #1: No

---

## [Author Response · Author response to Decision Letter 0]

17 Mar 2020

Dear Editor,

please find the response to your email in the uploaded cover letter and the reviewers comments in the letter called "Response to Reviewers"

Kind Regards,

Mirko Heckert

---

## [Editor Report · Decision Letter 1]

15 Apr 2020

Novel multi-energy X-ray imaging methods: Experimental results of new image processing techniques to improve material separation in computed tomography and direct radiography

PONE-D-19-24675R1

Dear Dr. Heckert,

We are pleased to inform you that your manuscript has been judged scientifically suitable for publication and will be formally accepted for publication once it complies with all outstanding technical requirements.

With kind regards,

David Fyhrie

Academic Editor

PLOS ONE
---

## [Editor Report · Acceptance letter]

20 Apr 2020

PONE-D-19-24675R1 

Novel multi-energy X-ray imaging methods: Experimental results of new image processing techniques to improve material separation in computed tomography and direct radiography 

Dear Dr. Heckert:

I am pleased to inform you that your manuscript has been deemed suitable for publication in PLOS ONE. Congratulations! Your manuscript is now with our production department. 

With kind regards,

on behalf of

Dr. David Fyhrie 

Academic Editor

PLOS ONE